Current approaches for executing big data science projects—a systematic literature review

Saltz Jeffrey S. 1
Krasteva Iva 2 iva.krasteva@gate-ai.eu
1 Syracuse University , Syracuse , NY, United States of America
2 GATE Institute, Sofia University , Sofia , Bulgaria
Piccolo Stephen
Electronic publication date: 2022 Feb 21
Publication date: 2022
Volume: 8
Electronic Location ID: e862
Received 2021 Sep 13; Accepted 2022 Jan 3
Copyright: © 2022 Saltz and Krasteva
Copyright year: 2022
Copyright holder: Saltz and Krasteva
License: This is an open access article distributed under the terms of the Creative Commons Attribution License, which permits unrestricted use, distribution, reproduction and adaptation in any medium and for any purpose provided that it is properly attributed. For attribution, the original author(s), title, publication source (PeerJ Computer Science) and either DOI or URL of the article must be cited.
License URL: https://creativecommons.org/licenses/by/4.0/

Keywords: Big data science, Project execution, Process frameworks, Big data science workflows, Agile data science

Funding: H2020 WIDESPREAD-2018-2020 TEAMING 857155 Operational Programme Science and Education BG05M2OP001-1.003-0002-C01 This research work has been supported by the GATE project, funded by the H2020 WIDESPREAD-2018-2020 TEAMING Phase 2 programme under grant agreement no. 857155 and by Operational Programme Science and Education for Smart Growth under Grant Agreement No. BG05M2OP001-1.003-0002-C01. The funders had no role in study design, data collection and analysis, decision to publish, or preparation of the manuscript.

==============================
There is an increasing number of big data science projects aiming to create value for organizations by improving decision making, streamlining costs or enhancing business processes. However, many of these projects fail to deliver the expected value. It has been observed that a key reason many data science projects don’t succeed is not technical in nature, but rather, the process aspect of the project. The lack of established and mature methodologies for executing data science projects has been frequently noted as a reason for these project failures. To help move the field forward, this study presents a systematic review of research focused on the adoption of big data science process frameworks. The goal of the review was to identify (1) the key themes, with respect to current research on how teams execute data science projects, (2) the most common approaches regarding how data science projects are organized, managed and coordinated, (3) the activities involved in a data science projects life cycle, and (4) the implications for future research in this field. In short, the review identified 68 primary studies thematically classified in six categories. Two of the themes (workflow and agility) accounted for approximately 80% of the identified studies. The findings regarding workflow approaches consist mainly of adaptations to CRISP-DM (vs entirely new proposed methodologies). With respect to agile approaches, most of the studies only explored the conceptual benefits of using an agile approach in a data science project (vs actually evaluating an agile framework being used in a data science context). Hence, one finding from this research is that future research should explore how to best achieve the theorized benefits of agility. Another finding is the need to explore how to efficiently combine workflow and agile frameworks within a data science context to achieve a more comprehensive approach for project execution.

Introduction

There is an increasing use of big data science across a range of organizations. This means that there is a growing number of big data science projects conducted by organizations. These projects aim to create value by improving decision making, streamlining costs or enhancing business processes.

However, many of these projects fail to deliver the expected value (Martinez, Viles & Olaizola, 2021). For example, VentureBeats (2019) noted that 87% of data science projects never make it into production and a NewVantage survey (NewVantage Partners, 2019) reported that for 77% of businesses, the adoption of big data and artificial intelligence (AI) initiatives is a big challenge. A systematic review over the grey and scientific literature has found 21 cases of failed big data projects reported over the last decade (Reggio & Astesiano, 2020). This is due, at least in part, to that fact that data science teams generally suffer from immature processes, often relying on trial-and-error and Ad Hoc processes (Bhardwaj et al., 2015; Gao, Koronios & Selle, 2015; Saltz & Shamshurin, 2015). In short, big data science projects often do not leverage well-defined process methodologies (Martinez, Viles & Olaizola, 2021; Saltz & Hotz, 2020). To further emphasize this point, in a survey to data scientists from both industry as well as from not-for-profit organizations, 82% of the respondents did not follow an explicit process methodology for developing data science projects, and equally important, 85% of the respondents stated that using an improved and more consistent process would produce more effective data science projects (Saltz et al., 2018).

While a literature review in 2016 did not identify any research focused on improving data science team processes (Saltz & Shamshurin, 2016), more recently, there has been increase in the studies specifically focused on how to organize and manage big data science projects in more efficient manner (e.g. Martinez, Viles & Olaizola, 2021; Saltz & Hotz, 2020).

With this in mind, this paper presents a systematic review of research focused on the adoption of big data science process frameworks. The purpose is to present an overview of research works, findings, as well as implications for research and practice. This is necessary to identify (1) the key themes, with respect to current research on how teams execute data science projects, (2) the most common approaches regarding how data science projects are organized, managed and coordinated, (3) the activities involved in a data science projects life cycle, and (4) the implications for future research in this field.

The rest of the paper is organized as follows: “Background and Related Work” section provides information on big data process frameworks and the key challenges with respect to teams executing big data science projects. In the “Survey Methodology” section, the adopted research methodology is discussed, while the “Results” section presents the findings of the study. The insights from this SLR as well as implications for future research and limitations of the study are highlighted in the “Discussion” section. “Conclusions” section concludes the paper.

Background and Related Work

It has been frequently noted that project management (PM) is a key challenge for successfully executing data science projects. In other words, a key reason many data science projects fail is not technical in nature, but rather, the process aspect of the project (Ponsard et al., 2017). Furthermore, Espinosa & Armour (2016) argue that task coordination is a major challenge for data projects. Likewise, Chen, Kazman & Haziyev (2016) conclude that coordination among business analysts, data scientists, system designers, development and operations is a major obstacle that compromises big data science initiatives. Angée et al. (2018) summarized the challenge by noting that it is important to use an appropriate process methodology, but which, if any, process is the most appropriate is not easy to know.

The importance of using a well-defined process framework

This data science process challenge, in terms of knowing what process framework to use for data science projects, is important because it has been observed that big data science projects are non-trivial and require well-defined processes (Angée et al., 2018). Furthermore, using a process model or methodology results in higher quality outcomes and avoids numerous problems that decrease the risk of failure in data analytics projects (Mariscal, Marbán & Fernández, 2010). Example problems that occur when a team does not use a process model include the team being slow to share information, deliver the wrong result, and in general, work inefficiently (Gao, Koronios & Selle, 2015; Chen et al., 2017).

The most common framework: CRISP-DM

The CRoss-Industry Standard Process for Data Mining (CRISP-DM) (Chapman et al., 2000) along with Knowledge Discovery in Databases (KDD) (Fayyad, Piatetky-Shapiro & Smyth, 1996), which both were created in the 1990s, are considered ‘canonical’ methodologies for most of the data mining and data science processes and methodologies (Martinez-Plumed et al., 2019; Mariscal, Marbán & Fernández, 2010). The evolution of those methodologies can be traced forward to more recent methodologies such as Refined Data Mining Process (Mariscal, Marbán & Fernández, 2010), IBM’s Foundational Methodology for Data Science (Rollins, 2015) and Microsoft’s Team Data Science Process (Microsoft, 2020).

However, recent surveys show that when data science teams do use a process, CRISP-DM has been consistently the most commonly used framework and de facto standard for analytics, data mining and data science projects (Martinez-Plumed et al., 2019; Saltz & Hotz, 2020). In fact, according to many opinion polls, CRISP-DM is the only process framework that is typically known by data science teams (Saltz, n.d.), with roughly half the respondents reporting to use some version of CRISP-DM.

Specifically, CRISP-DM defines the following six phases: Business understanding—includes identification of business objectives and data mining goals

Data understanding—involves data collection, exploration and validation

Data preparation—involves data cleaning, transformation and integration

Modelling—includes selecting modelling technique and creating and assessing models

Evaluation—evaluates the results against business objectives

Deployment—includes planning for deployment, monitoring and maintenance.

CRISP-DM allows some high-level iteration between the steps (Gao, Koronios & Selle, 2015). Typically, when a project uses CRISP-DM, the project moves from one phase (such as data understanding) to the next phase (e.g., data preparation). However, as the team deems appropriate, the team can go back to a previous phase. In a sense, one can think of CRISP-DM as a waterfall model for data mining (Gao, Koronios & Selle, 2015).

While CRISP-DM is popular, and CRISP-DM’s phased based approach is helpful to describe what the team should do, there are some limitations with the framework. For example, the framework provides little guidance on how to know when to loop back to a previous phase, iterate on the current phase, or move to the next phase. In addition, CRISP-DM does not contemplate the need for operational support after deployment.

The stated need for more research

Given that many data science teams do not use a well-defined process and that others use CRISP-DM with known challenges, it is not surprising that there has been a consistent calling for more research with respect to data science team process. For example, in Cao’s discussion of Data Science challenges and future directions (Cao & Fayyad, 2017), it was noted that one of the key challenges in analyzing data includes developing methodologies for data science teams. Gupte (2018) similarly noted that the best approach to execute data science projects must be studied. However, even with this noted challenge on data science process, there is a well-accepted view that not enough has been written about the solutions to tackle these problems (Martinez, Viles & Olaizola, 2021).

Is there still a need for more research?

This lack of research on data science process frameworks was certainly true 6 years ago, when the need for concise, thorough and validated information regarding the ways data science projects are organized, managed and coordinated was noted (Saltz, 2015). This need was further clarified when, in a literature review of big data science process research, no papers were found that focused on improving a data science team’s process or overall project management (Ransbotham, David & Prentice, 2015). This was also consistent with the view that most big data science research has focused on the technical capabilities required for data science and has overlooked the topic of managing data science projects (Saltz & Shamshurin, 2016).

However, much has happened during the past 6 years, with respect to research on data science process frameworks. With this in mind, to help move the field forward, this research aims to focus on the following research questions: RQ1: Has research in this domain increased recently?

RQ2: What are the most common approaches regarding how data science projects are organized, managed and coordinated?

RQ3: What are the phases or activities in a data science project life cycle?

Survey Methodology

While there are many approaches to a literature review, one approach, which is followed in this research, is to combine quantitative and qualitative analysis to provide deeper insights (Joseph et al., 2007). Furthermore, the systematic literature review conducted in this study leveraged the guidelines for performing SLRs suggested by Kitchenham & Charters (2007) and the data were collected in a similar manner as described in Saltz & Dewar (2019). Hence, the SLR process consisted of three phases: planning, conducting and reporting the review. The subsections below present the outcomes of the first two phases, while the results of the review are reported in the next section.

Planning the review

In general, systematic reviews address the need to summarize and present the existing information about some phenomenon in a thorough and unbiased manner (Kitchenham & Charters, 2007). As previously noted, the need for concise, thorough and validated information regarding the ways data science projects are organized, managed and coordinated is justified by the lack of established and mature methodologies for executing data science projects. This has led to our previously defined research questions, which are the drivers for how we structured our research.

The study search space comprises the following five online sources: ACM Digital Library, IEEEXplore, Scopus, ScienceDirect and Google Scholar. In addition to online sources, the search space might be enriched with reference lists from relevant primary studies and review articles (Kitchenham & Charters, 2007). Specifically, the papers that cite the study providing justification for the present research (Saltz, 2015) and the previous SLR on the subject (Saltz & Shamshurin, 2016) are added to the study search space.

Our search strategy includes both metadata and full-text searches over the selected online sources. The search phases that were identified after a couple of iterations, cover the two key concepts relevant to the study: Data science related terms: (“data science” OR “big data” OR “machine learning”).

Project execution related terms: (“process methodology” OR “team process” OR “team coordination” OR “project management”).

To determine whether a paper should be included in our analysis, the following selection criteria are defined:

Inclusion criteria: Papers that fully or partly include a description of the organization, management or coordination of big data science projects.

Papers that suggest specific approaches for executing big data science projects.

Papers that were published after 2015.

Exclusion criteria: Papers that are not written in English

Papers that did not focus on data science team process, but rather, focused on using data analytics to improve overall project management processes were excluded.

Papers that had no form of peer review (e.g. blogs).

Papers with irrelevant document type such as posters, conference summaries, etc.

Our exclusion of papers that discussed the use of analytics for overall project management considerations was driven by our desire to focus this research on understanding the specific attributes of data science projects, and how different frameworks were, or were not, applicable in the context of a data science project. This does not imply that data science has no role in helping to improve overall project management approaches. In fact, data science can and should add to the field of general project management, but we view this analysis as beyond the scope of our research.

The selection procedure describes how the selection criteria will be applied while conducting the study (Kitchenham & Charters, 2007; Saltz & Dewar, 2019). In our case, we planned two selection steps: Step1: Title and abstract screen—Initially, after the relevant papers from the search space are identified according to the study search strategy, the selection criteria will be applied considering only the title and the abstracts of the papers. This step is to be executed by the two authors over different sets of identified papers.

Step2: Full text screen—The full text of the candidate papers will then be reviewed by the two authors independently to identify the final set of primary studies to be included for further data analysis.

The approach for data extraction and synthesis followed in our study is based on the content analysis suggested in Elo & Kyngäs (2008), Hsieh & Shannon (2005). After exploring the key concepts used within each of the primary studies, general research themes are to be identified and further analysis of the data with respect to the study research questions is to be performed in both qualitative and quantitative manner.

Conducting the review

The SLR procedure was performed at the beginning of May, 2021. Because of the differences in running the searches over the online sources included in our search space, the identification of research and the first step of the selection procedure for Google Scholar were executed independently from the other digital libraries.

Three searches for the identification of relevant studies were executed over Google Scholar database with the following search strings: Search 1, the “data science” search: “data science” AND (“process methodology” OR “team process” OR “team coordination” OR “project management”).

Search 2, the “machine learning” search: “machine learning” AND (“process methodology” OR “team process” OR “team coordination” OR “project management”).

Search 3, the “big data” search: “big data” AND (“process methodology” OR “team process” OR “team coordination” OR “project management”).

Since the number of papers returned after executing the searches were very large, via a snowball sampling approach, only the first 220 papers in each result sets were included for further analysis. The first step of the selection procedure was executed for the unique papers in each of the sets and 48 papers were selected as candidates for primary studies. Table 1 shows the exact number of papers returned after running the searches and the first step of the selection procedure for Google Scholar.

Table 1 Retrieved and candidate papers from Google Scholar.

Search strings	Retrieved papers	Candidate papers	
“data science” search string	9,200 (first 220 used)	37	
“machine learning” search string	17,800 (first 220 used)	1	
“big data” search string	17,600 (first 220 used)	10	

Executing the initial search strings over the digital libraries resulted a vast number of papers (e.g., over 1,500 papers for IEEE Xplore full text). Motivated by the results of the executed searches in Google Scholar, an optimization of the search terms was introduced. Since the ratio of candidate to retrieved papers for the “machine learning” Google Scholar search string was very low and only one paper was selected after the first step of the selection procedure, we removed the term “machine learning” from the initial “Data science related terms” search phrase. The final search string that was used for identification of studies from the digital libraries the was: (“data science” OR “big data” OR “machine learning”) AND (“process methodology” OR “team process” OR “team coordination” OR “project management”).

Both metadata and full text searches were performed over the four digital libraries: ACM Digital Library—full text search.

IEEEXplore—metadata-based and full text searches.

Scopus—metadata-based search.

ScienceDirect—metadata-based search.

When executing the searches, appropriate filters helping to meet inclusion and exclusion criteria for each of the sources were applied where available. We used Mendeley as a reference management tool to help us organize the retrieved papers and to automate the removal of duplicates. A total of 1,944 was returned by the searches, from which 1,697 were unique papers. After executing the title and abstract screen, 98 papers were selected for candidates for primary studies. The exact numbers of retrieved and candidate papers are presented in Table 2. The numbers shown in the table include papers duplicated across the digital libraries.

Table 2 Retrieved and candidate papers from digital libraries.

Digital library search	Retrieved papers	Candidate papers	
Scopus: Metadata	327	52	
ACM: Full text	330	18	
IEEE: All metadata	197	24	
IEEE: Full Text	1,066	36	
Science Direct: Metadata	24	5	

The relevant studies search space comprised the papers that cite the two studies which provide the proper justification and relevant background for our research, namely (Saltz, 2015) and (Saltz & Shamshurin, 2016). A total of 159 papers were found to cite the two papers. After filtering the papers by screening the titles and abstracts, 64 of those papers were selected for candidate primary studies.

A consolidated list of all the candidate papers which were selected in the previous step of the selection procedure was created. The list included 120 unique papers. After performing the next step of the selection procedure (full text review), 68 papers were selected. These papers comprised the list of primary studies that were further analyzed to provide the answers to our research questions. The steps of the SLR procedure that led to the identification of the primary studies for our study are presented in Fig. 1.

Figure 1 Steps of the SLR procedure for identification of primary studies.

Following the guidelines by Cruzes & Dybå (2011), thematic analysis and synthesis was applied during data extraction and synthesis. We used the integrated approach (Cruzes & Dybå, 2011), which employs both inductive and deductive code development, for retrieving the research themes related to the execution of data science projects as well as for defining the categories of workflow approaches and the themes for agile adoption presented in the following section.

Results

This section presents the findings of the SLR with regard to the three research questions defined in the planning phase.

Research activity in this domain (RQ1)

As shown in Fig. 2, there has been an increase in the number of articles published over time. Note that the review was in done in May 2021, so the 2021 year was on pace to have more papers than any other year (i.e., over the full year, 2021 was on pace to have 18+ papers). Furthermore, it is likely that 2020 had a reduction due to COVID.

Figure 2 Number of papers per year.

We also explored publishing outlets. Specifically, Fig. 3 shows the number of papers for each publisher. IEEE was the most frequent publisher, with 31 (46%) papers, due in part to a yearly IEEE workshop on this domain, that started in 2015. The next highest publisher was ACM, with nine papers (13%).

Figure 3 Number of papers for each publisher.

Approaches for executing data science projects (RQ2)

Table 3 provides an overview of the six themes identified, with respect to the approaches for defining and using a data science process framework. The table also shows the relevant primary studies. While the six themes that we identified in our SLR are all relevant to project execution, there was a wide range in the number of papers published for the different themes. The ratio of publications across the different themes provides a high-level view of current research efforts regarding the execution of data science projects.

Table 3 Themes relevant to execution of data science projects.

Theme	Primary studies	Total number	
Workflows	See Table 4	27	
Agility	See Table 5	26	
Process adoption	(Saltz, 2017, 2018; Saltz & Hotz, 2021; Soukaina et al., 2019; Saltz & Shamshurin, 2017; Shamshurin & Saltz, 2019a)	6	
General PM	(Saltz & Shamshurin, 2015; Mao et al., 2019; Ramesh & Ramakrishna, 2018; Mullarkey et al., 2019)	4	
Tools	(Marin, 2019; Wang et al., 2019; Chen et al., 2020; Saltz et al., 2020; Crowston et al., 2021)	5	
Reviews	(Saltz, 2015; Saltz & Shamshurin, 2016; Schröer, Kruse & Gómez, 2021; Plotnikova, Dumas & Milani, 2020; Krasteva & Ilieva, 2020; Martinez, Viles & Olaizola, 2021; Saltz et al., 2018)	7	

Table 4 Workflow categories.

Category	Reference workflows	Primary studies	
New	N\A	(Dutta & Bose, 2015; Shah, Gochtovtt & Baldini, 2019)	
CRISP-DM	(Grady, 2016; Grady, Payne & Parker, 2017; Ahmed, Dannhauser & Philip, 2019)	
KDD, CRISP-DM	(Amershi et al., 2019)	
Standard	CRISP-DM	(Saltz, Shamshurin & Crowston, 2017; Saltz, Heckman & Shamshurin, 2017; Saltz & Heckman, 2018)	
Specialization	CRISP-DM	(Kalgotra & Sharda, 2016; Schwenzfeier & Gruhn, 2018)	
KDD	(Vernickel et al., 2019)	
Extension	CRISP-DM	(Ponsard, Touzani & Majchrowski, 2017; Ponsard et al., 2017; Asamoah & Sharda, 2019; Qadadeh & Abdallah, 2020)	
KDD	(Silva, Saraee & Saraee, 2019)	
other	(Lin & Huang, 2017; Angée et al., 2018; Baijens & Helms, 2019)	
Enrichment	CRISP-DM	(Yamada & Peran, 2017; Martinez-Plumed et al., 2019; Kolyshkina & Simoff, 2019; Costa & Aparicio, 2020; Kordon, 2020; Fahse, Huber & van Giffen, 2021)	
other	(Zhang, Muller & Wang, 2020)	

Table 5 Agility themes.

Theme	Primary studies	Type	Total number	
Conceptual Benefits of Agility	(Franková, Drahošová & Balco, 2016; Dharmapal & Sikamani, 2016; Grady, Payne & Parker, 2017; Al-Jaroodi, Hollein & Mohamed, 2017; Ponsard, Touzani & Majchrowski, 2017; Becker, 2017; Ponsard et al., 2017; Hassani, El Idrissi & Abouabdellah, 2018; Demigha, 2019; Shah, Gochtovtt & Baldini, 2019; Saltz & Suthrland, 2019; Reggio & Astesiano, 2020; Baijens, Helms & Kusters, 2020; Aho et al., 2020; Rotondo & Quilligan, 2020)	Conceptual	15 (58%)	
Challenges in Scrum	(Saltz, Shamshurin & Crowston, 2017; Saltz, Heckman & Shamshurin, 2017; Singla, Bose & Naik, 2018; Saltz & Shamshurin, 2019; Baijens, Helms & Iren, 2020)	Case Study	5 (19%)	
Scrum is used	(Maria et al., 2015; Saltz & Hotz, 2020)	Case Study	2 (7%)	
Conceptual Benefits of Scrum	(Larson & Chang, 2016; Dabrowski, 2021)	Conceptual	2 (7%)	
Conceptual Benefits of Lean	(Ahmed, Dannhauser & Philip, 2019)	Conceptual	1 (4%)	
Challenges in Kanban	(Shamshurin & Saltz, 2019b)	Case Study	1 (4%)	

Below we provide a description for each of the themes, with an expanded focus on the two most popular themes (workflows and agility).

Workflows papers explored how data science projects were organized with respect to the phases, steps, activities and tasks of the execution process (e.g., CRISP-DM’s project phases). There were 27 papers in this theme, which is about 40% of the total number of primary studies. Workflow approaches are discussed in our second research question and a detailed overview of the relevant studies will be provided in the following section.

Agility papers described the adoption of agile approaches and considered specific aspects of project execution such as the need for iterations or how teams should coordination and collaborate. The high number of papers categorized in the Agility theme (26 out of 68) might be due to the successful adoption of agile methodologies in various software development projects. The theme will be covered in the next section since agile adoption is also relevant to our second research question. Seven papers explored both the workflows and agility themes.

Process adoption papers discussed the key factors as well as the challenges for a data science team to adopt a new process. Specifically, the papers that discussed process adoption considered questions such as acceptance factors (Saltz, 2017, 2018; Saltz & Hotz, 2021), project success factors (Soukaina et al., 2019), exploring the application of software engineering practices in the data science context (Saltz & Shamshurin, 2017), and would deep learning impact a data science teams process adoption (Shamshurin & Saltz, 2019a).

General PM papers discussed general project management challenges. These papers did not focus on addressing any data science unique characteristics, but rather, general management challenges such as the team’s process maturity (Saltz & Shamshurin, 2015), the need for collaboration (Mao et al., 2019), the organizational needs and challenges when executing projects (Ramesh & Ramakrishna, 2018) and training of human resources (Mullarkey et al., 2019).

Tools focused papers described new tools that could improve the data science team’s productivity. Five papers explored how different tools, both custom and commercial, could be used to support various aspects of the execution of the data science projects. The tools explored focused on communication and collaboration (Marin, 2019; Wang et al., 2019), Continuous Integration/Continuous Development (Chen et al., 2020), the maintainability of a data science project (Saltz et al., 2020) and a tool to improve the coordination of the data science team (Crowston et al., 2021).

Reviews were papers that reported on a SLR for a specific topic related to data science project execution or papers that report on an industry survey. An SLR aiming to find out benefits and challenges on applying CRISP-DM in research studies is presented in Schröer, Kruse & Gómez (2021). How different data mining methodologies are adapted in practice is investigated in Plotnikova, Dumas & Milani (2020). That literature review covered 207 peer-reviewed and ‘grey’ publications and identified four adaptation patters and two recurrent purposes for adaptation. Another SLR focused on experience reports and explored the adoption of agile software development methods in data science projects (Krasteva & Ilieva, 2020). An extensive critical review over 19 data science methodologies is presented in Martinez, Viles & Olaizola (2021). The paper also proposed principles of an integral methodology for data science which should include the three foundation stones: project, team and data & information management. Professionals with different roles across multiple organizations were surveyed in Saltz et al. (2018) about the methodology they used in their data science projects and whether an improved project management process would benefit their results. The two papers that formed the core of our search space of related papers (Saltz, 2015) and (Saltz & Shamshurin, 2016), were also included in the Reviews thematic category.

Workflow approaches

The thematic analysis of the workflows for data science projects revealed that the workflows might be broadly categorized in three groups: (1) standard, (2) new, and (3) adapted workflows. Furthermore, three sub-categories of adapted workflows were synthesized based on the aim of the adoption: Specialization—adjustments to standard workflows, which are made to better suit particular big data technology or specific domain.

Extension—addition of new steps, tasks or activities to extend standard workflow phases.

Enrichment—extension of the scope of a standard workflow to provide more comprehensive coverage of the project execution activities.

An overview of workflow categories and respective primary studies is presented in Table 4. Multiple studies of the same workflow are shown in brackets. Most of the workflows use a standard framework as a reference point for specification of both new and adapted workflows. As seen in Table 4, CRISP-DM provides the basis for the majority of the workflow papers. Below we explore each of these categories in more depth.

New workflows

While the workflow proposed in Grady (2016) make use of CRISP-DM activities, a new workflow with four phases, five stages and more than 15 activities was designed to accommodate big data technologies and data science activities. Providing a more focused technology perspective (Amershi et al., 2019) proposes a nine-stage workflow for integrating machine learning into application and platform development. Uniting the advantages of experimentation and iterative working along with a greater understanding of the user requirements, a novel approach for data projects is proposed in Ahmed, Dannhauser & Philip (2019). The suggested workflow consists of three stages and seven steps and integrates the principles of the Lean Start-up method and design thinking with CRISP-DM activities. The workflows in Dutta & Bose (2015) and Shah, Gochtovtt & Baldini (2019) are designed and used in companies, and integrate strategic perspective with planning, management and implementation.

Standard workflows

Three of the primary studies reported on using CRISP-DM in student projects and compared and contracted the adoption of different methodologies (e.g. CRISP-DM, Scrum and Kanban) for executing data science projects.

Workflow specializations

Specialization category is the smallest of the three adaption sub-categories. Two of the workflows in this category were based on CRISP-DM and were specialized for sequence analysis (Kalgotra & Sharda, 2016) or anomaly detection (Schwenzfeier & Gruhn, 2018). In addition, a revised KDD procedure model for time-series data was proposed in Vernickel et al. (2019).

Workflow extensions

An extension to CRISP-DM for knowledge discovery on social networks was specified as a seven-stage workflow that can be applied in different domains intersecting with social network platforms (Asamoah & Sharda, 2019). While this workflow extended CRISP-DM for big data, the workflows in Ponsard, Touzani & Majchrowski (2017) and Qadadeh & Abdallah (2020) added additional workflow steps focused on identification of data value and business objectives. An extension to KDD for public healthcare was proposed in Silva, Saraee & Saraee (2019). The suggested workflow implies user-friendly techniques and tools to help healthcare professionals use data science in their daily work. By performing a SLR of recent developments in KD process models (Baijens & Helms, 2019) proposes relevant adjustments of the steps and tasks of the Refined Data Mining Process (Mariscal, Marbán & Fernández, 2010). The IBM’s Analytics Solutions Unified Method for Data Mining/predictive analytics (ASUM-DM) is extended in Angée et al. (2018) for a specific use case in the banking sector with focus on big data analytics, prototyping and evaluation. A software engineering lifecycle process for big data projects is proposed in Lin & Huang (2017) as an extension to the ISO/IEC standard 15288:2008.

Workflow enrichments

There were several papers that extend CRISP-DM in different dimensions. The studies in Kolyshkina & Simoff (2019) and Fahse, Huber & van Giffen (2021) addressed two important aspects of ML solutions—interpretability and bias, respectively. They suggested new activities and methods integrated in CRISP-DM steps for satisfying desired interpretability level and for bias prevention and mitigation. A novel approach for custom workflow creation from a flexible and comprehensive Data Science Trajectory map of activities was suggested in Martinez-Plumed et al. (2019). The approach is designed to address the diversity of data science projects and their exploratory nature. The workflow presented in Kordon (2020) proposes improvements to CRISP-DM in several areas—maintenance and support, knowledge acquisition and project management. Scheduling, roles and tools are integrated with CRISP-DM in a methodology, presented in Costa & Aparicio (2020). Checkpoints and synchronization are used in the proposed in Yamada & Peran (2017) Analytics Governance Framework to facilitate communication and coordination between the client and the data science team. Collaboration is the primary focus in Zhang, Muller & Wang (2020), in which a basic workflow is extended with collaborative practices, roles and tools.

Agile approaches

As shown in Table 5, there were 26 papers that focused on the need for agility within data science projects. Only 31% of the papers actually reported on teams using an agile approach. The rest of the papers, 69% (18 of the 26 papers), were conceptual in nature. These conceptual papers explained why it makes sense that a framework should be helpful for a data science project but provided no examples that the framework actually helps a data science team.

Specifically, the vast majority of the papers (15 papers), explored the potential benefits of agility for data science projects. These papers were labeled general agility papers since they did not explicitly support any specific agile approach, but rather, noted the benefits teams should get by adopting an agile framework. The expected benefits of agility typically focused on the need for multiple iterations to support the exploratory nature of data science projects, especially since the outcomes are uncertain. This would allow teams to adjust their future plans based on the results of their current iteration.

Two papers discussed the potential benefits of Scrum. However, five papers reported on the difficulty teams encountered when they actually tried to use Scrum. Often times, issues arose due to the challenge in accurately estimating how long a task would take to complete. This issue of task estimation impacted the team’s ability to determine what work items could fit into a sprint. Two other papers reported on the use of Scrum within data science team, but both of those papers did not describe in depth how the team used Scrum, nor if there were any benefits or issues due to their use of Scrum.

Finally, one paper discussed the conceptual benefits of using a lean approach and a different paper reported on the challenge in using Kanban (which can be thought as supporting both agility and lean principles). That paper explored the need for the process master role, similar to the Scrum Master role in Scrum.

Combined approaches

The seven papers that covered both the workflow and agility themes presented a more comprehensive methodology for project execution. Several proposed new frameworks (Grady, Payne & Parker, 2017; Ponsard, Touzani & Majchrowski, 2017; Ponsard et al., 2017; Ahmed, Dannhauser & Philip, 2019). All of the newly proposed frameworks defined a new workflow (typically based on CRISP-DM), and also suggested that the project do iterations and focus on creating a minimal viable product (MVP). However, there was no consensus on if the iterations should be time-boxed or capability based. Furthermore, there no consensus on how to integrate the data science life cycle into each iteration. In fact, two papers didn’t explicitly address this question (Ponsard, Touzani & Majchrowski, 2017; Ponsard et al., 2017) and another article implied that something should be done for each phase in each sprint (Grady, Payne & Parker, 2017). Yet another article suggested that maybe some iterations focus on a specific phase and other iterations might focus on more than one phase (Ahmed, Dannhauser & Philip, 2019).

Three articles analyzed existing frameworks, including both workflow and agile frameworks (Saltz, Shamshurin & Crowston, 2017; Saltz, Heckman & Shamshurin, 2017; Shah, Gochtovtt & Baldini, 2019). For both of these articles, there was not explicit discussion on how to integrate workflow frameworks with agile frameworks.

Data science project life cycle activities (RQ3)

Table 6 shows a synthesized overview of the life cycle phases mentioned in the workflow papers, presented above. This table also shows the number (and percentage) of papers that mention a specific data science life cycle phase. One can note that the most common phases are the CRISP-DM phases.

Table 6 Data science life cycle activities.

Theme	Total number	CRISP-DM phase	
Readiness assessment	1 (4%)		
Project organization	5 (18%)		
Business understanding	19 (68%)	✓	
Problem identification	8 (29%)		
Data acquisition	10 (36%)		
Data understanding	15 (54%)	✓	
Data preparation	21 (75%)	✓	
Feature engineering	4 (14%)		
Data analysis/Exploration	9 (32%)		
Modeling	25 (89%)	✓	
Model refinement	2 (7%)		
Evaluation	23 (82%)	✓	
Interpret/Explain	2 (7%)		
Deployment	20 (71%)	✓	
Business value	5 (18%)		
Monitoring	2 (7%)		
Maintenance	3 (11%)		

Discussion

The section presents further analysis on the findings of the study, highlighting the insights and implications for future research as well as exploring several validity threats.

Insights and implications for future research

The analysis of the information extracted for each primary study provided interesting insights on how data science projects are currently organized, managed and executed. The findings regarding categories of workflows confirm the trend observed in Plotnikova, Dumas & Milani (2020) of the large number of adaptations of workflow frameworks (vs proposing new methodologies). While CRISP-DM is reported to be the most widely used framework for data science projects (e.g. Saltz & Hotz, 2020), the adaptions of CRISP-DM in data science projects are much more commonly reported in the research literature, which raises the question if teams are adapting CRISP-DM, when they are using it within their project.

Most of the agility papers were conceptual in nature, and many of the other papers reported on issues when using Scrum. Hence, more research is needed to explore how to achieve the theorized benefits of agility, perhaps by adapting Scrum or using a different framework.

Combining workflow approaches with agile frameworks within a data science context is a way to achieve an integral framework for project execution. However, more research is needed on how to combine these two approaches. For example, the research presented in Martinez, Viles & Olaizola (2021) over the 19 methodologies for data science projects determined that only four of them could be classified as integral according to the criteria defined in the study. Specifying new data science methodologies that cover different aspects of project execution (e.g. team coordination, data and system engineering, stakeholder collaboration) is a promising direction for future research.

To explore if the life cycle activities mentioned in the workflow papers have changed over time, we conducted a comparative analysis with a similar SLR in which 23 data mining process models are compared based on process steps (Rotondo & Quilligan, 2020). As all of the papers from the previous SLR were prior to 2018, comparing the two SLR’s provides a way to see if the usage of different phases has changed over time. It was observed that the use of an exploratory phase (Data Analysis/Exploration) was increasing, while the model interpretation and explanation phase (Interpret/Explain) was decreasing. The last is perhaps due to these tasks being integrated into the evaluation phase.

Validity threats

Several limitations of the study present potential threats to its validity. One limitation is that the SLR was based on a specific set of search strings. It is possible a different search string could have identified other interesting articles. Adding an additional search space based on citations of relevant studies tried to mitigate the impact of this potential threat.

Another limitation is that while authors explored ACM Digital Library, IEEEXplore, Scopus, ScienceDirect and Google Scholar databases, which index high impact journals and conference papers from IEEE, ACM, SpringerLink, and Elsevier, it is possible that some relevant articles from other publication outlets could have been missed. In addition, the grey literature was not analyzed. This literature could have provided additional insights on the adoption of data science approaches in industrial settings. Yet another limitation is that the analysis and synthesis were based on qualitative content analysis and thematic synthesis of the selected articles by the research team. The authors tried to minimize the subjectivity of researchers’ interpretation by cross-checking papers to reduce bias.

Conclusions

This study presents a systematic review of research focused on the adoption of big data science process frameworks. The study shows that research on how data science projects are organized, managed and executed has increased significantly during the last 6 years. Furthermore, the review identified 68 primary studies and thematically classified these studies in six key themes, with respect to current research on how teams execute data science projects (workflows, agility, process adoption, general PM, tools, and reviews). CRISP-DM was the most common workflow discussed, and the different adaption patterns of CRISP-DM—specializations, extensions and enrichments, were the most common approaches for specifying and using adjusted workflows for data science projects.

However, standardized approaches explicitly designed for the data science context were not identified, and hence, is a gap in current research and practice. Similarly, with respect to agile approaches, more research is needed to explore how and if the conceptual benefits of agility noted in many of the identified papers can actually be achieved in practice. In addition, another direction for future research is to explore combining workflow and agile approaches into a more comprehensive framework that covers different aspects of project execution.

The current study can be enhanced and extended in three directions. First, the search space could be expanded by using the snowballing technique (Wohlin, 2014) for identification of relevant articles. Some of the primary studies identified in the current study can be used as seed papers in a future execution of the procedure. Second, conducting a multivocal literature review (Garousi, Felderer & Mäntylä, 2016) including grey literature can complement the results of the study by collecting more experience reports and real-world adoptions from industry. Finally, future research could explore if the process used should vary based on different industries, or if, the appropriate data science process is independent of the specific industry project context.

Additional Information and Declarations

Competing Interests

Author Contributions

Data Availability

The authors declare that they have no competing interests.

Jeffrey S. Saltz conceived and designed the experiments, performed the experiments, analyzed the data, prepared figures and/or tables, authored or reviewed drafts of the paper, and approved the final draft.

Iva Krasteva conceived and designed the experiments, performed the experiments, analyzed the data, prepared figures and/or tables, authored or reviewed drafts of the paper, and approved the final draft.

The following information was supplied regarding data availability:

All data (papers) reviewed and analysed are available in ACM Digital Library, IEEEXplore, Scopus, ScienceDirect and Google Scholar.

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
