# Peer review of "Current approaches for executing big data science projects—a systematic literature review"

_PeerJ Computer Science, doi:10.7717/peerj-cs.862_

## Round 0.1 · original submission · Minor Revisions

The reviewers appreciated this article and made some positive comments. They also listed some minor changes that they would like to see. Please consider their recommendations and submit a revision accordingly.

Reviewer 1 ·

Basic reporting

The manuscript presents a systematic review of frameworks for executing big data science projects. The contributions of the article are well presented. The manuscript concisely summarizes the current state of the art in data science project management frameworks. Furthermore, the findings and conclusions derived from the systematic review are of interest to the research community on organizational and management frameworks for big data science projects.

The structure of the article is correct, and there are sufficient background details and literature references provided. The article introduces the topic adequately. The lack of established and mature methodologies justifies the need for information about how data science projects are organized, managed, and coordinated. Thus, it adequately justifies the need for a systematic review in this field.

The paper is well written and well organized, uses professional, clear, and unambiguous English. Figures and tables in general are comprehensive and helpful.

Experimental design

The content of the article is consistent with the journal's Aims and Scope. Furthermore, the methodology for the systematic review is described with sufficient detail: both the search space, the search strategy, and the selection criteria & procedure are well justified and described to be reproducible by another investigator (given access to the mentioned digital libraries).
The references are correctly cited. The review is also organized logically into coherent paragraphs.

The reviewer noted a possible mismatch in the total number of papers:

- In Figure 1, in the Google Scholar search space, the number of retrieved studies is 37600. However, the total number of retrieved papers in Table 1 is 44600 (9200 + 17800 + 17600). Is this mismatch due to repeated articles? This difference should be clarified.

- In line 262, it is mentioned that "After executing the title and abstract screen, 98 papers were selected for candidates for primary studies.". The same number of articles (98) appears in Figure 1. However, the total number of candidate papers in table 2 is 135 (52+18+24+36+5). Again, is this mismatch due to repeated articles on different digital libraries? This difference should be clarified.

Validity of the findings

The novelty of the study is assessed in the manuscript. The conclusions are appropriately stated, connected to the three research questions, and are supported by the results. The conclusion identifies very interesting future directions for research in the field. The limitations of the study are also presented in the Discussion section.

- Lines 299-304: "While the six themes that we identified in our SLR are all relevant to project execution, there was a wide range in the number of papers published for the different themes. Exploring this ratio of publications across the different themes provides a high-level view of research activity for the different themes. In other words, the number of articles for a particular theme is indicative of the current focus, with respect to current research efforts for that theme regarding the execution of data science projects."
The reviewer found this paragraph not straightforward and difficult to read, with the same idea repeated twice. Therefore, the reviewer recommends simplifying this paragraph and making it easier to understand.

Additional comments

Typography errors:
- Lines 511-513: "Another limitation is that while authors explored ACM Digital Library, IEEEXplore, Scopus, ScienceDirect, and Google Scholar databases, which index high impact journals and conference papers from IEEE, ACM, SpringerLink, and Elsevier to identify all possible relevant articles.": The while clause has not been appropriately used; there is no second part to the sentence.
- Line 387: the text should be in bold
- Caption Figure 1: csteps => steps
- Table 4: There are unnecessary parentheses on Extension / CRISP-DM / Primary Studies

Reviewer 2 ·

Basic reporting

This is a very interesting piece of research.
The use of English was clear, professional, and unambiguous.
The review of literature is carried out following a logical approach and a consistent methodology. Reasonable exclusions are adequately justified.
The structure of the paper is good, however, Table 3 shows missing references in the first two lines (Error displayed)
The article is aligned with the aims and scope of the journal and potentially impactful as its novelty is identified in a broad review (based on recent and up to date research) of the approaches for executing big data science projects. However, the article is generically assessing the methods and approaches underestimating the importance (and relevance) of the industries in which big data can be applied.
The introduction adequately presents the subject and it supports enough the research rationale and scope.

Experimental design

Rigorous investigation, excellent description of the methods and appropriate citations.

Validity of the findings

The validity of the findings is supported by adequate and complete methodology.
I encourage the authors to further investigate the industries (sectors) where (and why) the big data can be better applied.
Conclusions are well stated, however, they can be more developed. (Gaps and limitations adequately identified)

---

## Round 0.2 · accepted · Accept

Thank you for addressing the reviewers' comments. This paper will be a nice addition to the literature.